# Seasonal variation in dragonfly assemblage colouration suggests a link between thermal melanism and phenology

Roberto Novella-Fernandez [1] ✉, Roland Brandl[2], Stefan Pinkert [3], Dirk Zeuss [4] & Christian Hof [1,5]

Phenology, the seasonal timing of life events, is an essential component of diversity patterns. However, the mechanisms involved are complex and understudied. Body colour may be an important factor, because dark-bodied species absorb more solar radiation, which is predicted by the Thermal Melanism Hypothesis to enable them to thermoregulate successfully in cooler temperatures. Here we show that colour lightness of dragonfly assemblages varies in response to seasonal changes in solar radiation, with darker early- and late-season assemblages and lighter mid-season assemblages. This finding suggests a link between colour-based thermoregulation and insect phenology. We also show that the phenological pattern of dragonfly colour lightness advanced over the last decades. We suggest that changing seasonal temperature patterns due to global warming together with the static nature of solar radiation may drive dragonfly flight periods to suboptimal seasonal conditions. Our findings open a research avenue for a more mechanistic understanding of phenology and spatio-phenological impacts of climate warming on insects.

Improving our capacity to predict biodiversity changes requires a profound understanding of the mechanisms that drive the variation of life in space and time. Over the last decades, ecological research has greatly advanced in documenting spatial patterns of diversity variation across large scales (e.g.[1,2]) as well as their underlying environmental drivers[3–5]. Besides spatial patterns, many taxa show characteristic seasonal replacement of species caused by species´ particular timing of their life events, i.e. phenologies. The mechanisms driving phenological diversity patterns are, compared to those driving spatial diversity, much more complex and poorly understood, which critically limits conclusions about the spatio-temporal changes that underpin species responses to climate change[6].

Insects constitute the vast majority of terrestrial animal species. In temperate regions, many insect groups show seasonal replacement of species[7] resulting from species-specific phenologies. Insect phenologies are the outcome of the interaction between seasonal developmental constraints and environmental life cycle regulation[8–10], in which life cycles are synchronised to optimal seasonal moments where fitness is maximized—a process named phenological fundamental tracking[11]. Within this fundamental tracking system, phenological events such as adult emergence are triggered by environmental cues that link to subjacent drivers of optimal timing[11] (Fig. 1a). For instance, butterfly emergence may be triggered by certain temperature and photoperiod levels[8] that align to the seasonal appearance of their host plants[12]. While most phenological research has so far focused on describing species-specific phenological events in response to environmental factors[13], little is understood about the underlying drivers of optimal timing[6,14]. Moreover, phenological shifts constitute one of the

[1]Technical University of Munich, Terrestrial Ecology Research Group, Department for Life Science Systems, School of Life Sciences, Freising, Germany. [2]Department of Ecology—Animal Ecology, Philipps-University Marburg, Marburg, Germany. [3]Department of Conservation Ecology, Philipps-Universität Marburg, Marburg, Germany. [4]Department of Geography—Environmental Informatics, Philipps-Universität Marburg, Marburg, Germany. [5]Department of Global Change Ecology, Biocentre, Julius-Maximilians-Universität Würzburg, Würzburg, Germany. ✉e-mail: r.novellaf@outlook.com

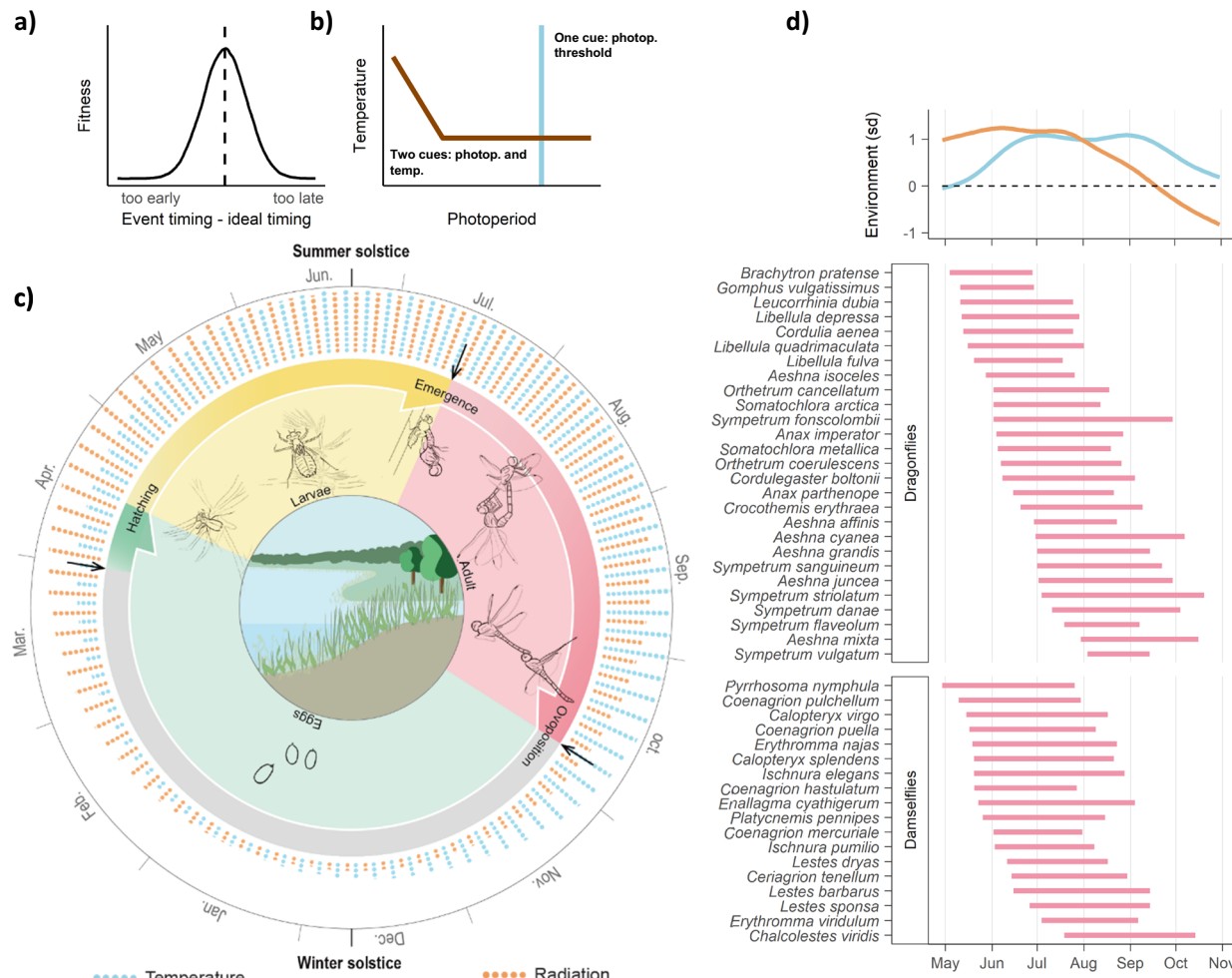

**Fig. 1 | Phenology of Odonata in relation to seasonal environmental conditions.**
**a**, **b** Phenological fundamental tracking regulates species´ life cycles by synchronising them to optimal seasonal moments. **a** Fitness increases when the timing of phenological events aligns to the timing of ideal environmental conditions.
**b** Phenological responses are triggered by environmental cues, such as certain photoperiod threshold (blue line), or combined photoperiod and temperature thresholds (brown line). These environmental cues link to−unknown−underlying drivers of optimal timing (a and b adapted from[11]). **c** Example of the life cycle of the dragonfly *Sympetrum striolatum* which is regulated by seasonal environmental conditions. Arrows note phenological events triggered by environmental cues. Eggs are oviposited in October when cold temperatures induce a diapause in their embryonic development that lasts until spring, when development resumes triggered by photoperiod and temperature cues[74]. Larvae hatch and develop for two to four months, depending on water temperature. Last instar larvae emerge, again triggered by photoperiod and temperature cues. Adult complete maturation and reproduce[74]. Alternatively, if oviposition occurs earlier, eggs can develop without diapause and larvae hatch in autumn to enter a diapause that lasts until spring, when development resumes[74]. Other temperate odonates have different life cycles varying in length from less than a year to several years[74], most of which are regulated by phenological fundamental tracking. **d** 5th and 95th percentile of dragonfly and damselfly flight periods in Great Britain, together with variation in radiation (orange) and temperature (blue) across the season (upper panel), indicated as standard deviation from the cross-year average (grey dashed line). *Coenagrion scitulum* is not shown because its rarity in the study system does not allow to derive a representative flight period. Artwork in (**c**) by Zijing Deng.

most obvious effects of climate change[15–17] but we do not understand their consequences for species, which could range from positive to negative depending on whether they can track shifting optimal seasonal conditions[18]. A failure to recognise the mechanisms underlying phenological patterns of diversity is a key knowledge gap in understanding the impact of climate change on natural systems[9,14,19].

Thermoregulation is a crucial mechanism regulating the life cycles and occurrences of ectotherms[20]. The Thermal Melanism Hypothesis (TMH) proposes that body colour darkness affects species´ thermoregulatory performance by increasing the absorption of solar radiation[21]. As a result, darker individuals and species are predicted to be able to occur in colder environments than their light-coloured counterparts[22]. TMH is well supported based on patterns of species' geographic distributions and community composition in a broad range of ectothermic taxa, including reptiles[23] and insects such as ants[24], Lepidoptera[25,26], and dragonflies and damselflies[27,28].

Although solar radiation and temperature vary seasonally as well as geographically, the relationship between thermal melanism and insect phenology has not been assessed, to our knowledge.

Here we test whether phenological patterns of insect flight periods vary with insect colour and seasonal environmental conditions as it would be predicted by thermal melanism. We use Odonata (suborders dragonflies and damselflies) as our study system because of the rich natural history record[29] and because it is one of the groups where the TMH has been most strongly supported, for instance driving functional community assembly across Europe and North America[27,28]. The strength of thermal melanism in this warm-adapted[30] group is related to their high thermoregulatory requirements, as odonates rely on−energetically costly−flight for all their essential activities (displacement, foraging, reproduction), for which particularly the larger-bodied dragonflies require thoracic temperatures above ambient levels (e.g. 27–36 °C[31,32]). In temperate latitudes, odonates show

characteristic phenological replacement of species´ flight periods (Fig. 1d) resulting from environmental regulation of species´ life cycles based on photoperiod and temperature cues together with developmental constraints (Fig. 1c). Whether certain underlying environmental drivers may determine optimal timing of their flight periods remains to be understood.

Recently available massive observational data allowed us to study odonate´s spatio-phenological diversity patterns in extensive detail and extent. We downloaded a database of over one million odonate records for Great Britain[33]. After grouping observations within fine spatio-phenological units and controlling for sampling effort by using rarefaction curves, we obtained datasets of 8159 and 4134 ecologically meaningful assemblages of dragonflies and damselflies, respectively, between May and October from 1990 to 2020 (see Methods for details). Values of body colour lightness per species, therefore assumed static, were obtained from scientific illustrations[34], see[27,28]. To quantify assemblage-level body colour lightness, we used community-weighted means[35], whose deviations from null expectations were then analysed in relation to the seasonal variation of the thermal environment. We also accounted for potential effects of phylogenetic relatedness as well as spatial autocorrelation (see Methods and Supplementary Figs. S3 and S5). We finally assessed changes in the phenological pattern of colour lightness over the last 30 years to evaluate potential shifts in response to climate change. We expected that the same mechanisms driving ectotherm spatial diversity patterns also contribute to determining insect flight periods. Specifically, we expected (1) colour lightness of odonate assemblages to follow predictions of the TMH and show phenological variation in response to seasonal changes in solar radiation intensity and temperature. Furthermore, we expected (2) thermal melanism responses to be stronger in the larger dragonflies than in damselflies, based on their higher thermoregulatory requirements[31,32,36]. Following reported phenological advances in odonate flight periods[37,38], we expected (3) to see advances in the phenological pattern of colour lightness over the last decades.

Our analyses show that the phenological variation of colour lightness of dragonfly, but not of damselfly assemblages follows the seasonal pattern of solar radiation, suggesting that colour-based thermoregulation and phenology in dragonflies are coupled. Furthermore, our findings indicate an advance of the phenological pattern of dragonfly colour lightness over the last 30 years, presumably associated with global warming.

## Results
### Colour lightness of dragonfly assemblages follows seasonal radiation
We found that colour lightness of dragonfly assemblages (CL) varied as expected—both phenologically and with latitude (4th degree polynomial model: $n = 8159$, $F_{5,8153} = 1147$, $R^2 = 0.41$, $P < 0.001$; Fig. 2a, d). CL decreased linearly with latitude as predicted by the THM (Lat: t =−29.66, $P < 0.001$. Fig. 2c), but the phenological effect was much stronger, with most explained variance (hierarchical partitioning: 84.8%) depending on the day of the year (Day: t = 15.55, $P < 0.001$; Day²: t =−13.8, $P < 0.001$; Day³: t = 11.92, $P < 0.001$; Day⁴: t =−10.11, $P < 0.001$; Figs. 2e). CL increased from May until mid-June to early July, and then gradually decreased until the end of August from where assemblages remained constantly dark until the end of the season in October (Fig. 2d, e). The phenological pattern of dragonfly CL was consistent when applying spatially restricted null models (Supplementary Fig. S1) which allows isolating the phenological component (see Methods for details). In contrast to dragonflies, CL of damselfly assemblages did not show latitudinal nor phenological patterns (Linear regression: $n = 4134$, $F_{2,4131} = 1.5$, $P = 0.22$; Fig. 2b, Supplementary Fig. S2). The observed spatio-phenological patterns of CL of both dragonflies and damselflies were robust to potentially confounding effects of phylogenetic non-

independence of traits (see methods and Supplementary Fig. S3). Variation in CL of dragonfly assemblages followed expectations from the TMH as CL increased non-linearly with the solar radiation received (Fig. 3) (2nd degree polynomial model: n = 7901, $F_{2,7898} = 1789$, $R^2 = 0.31$, $P < 0.001$; radiation: t =−14.92, $P < 0.001$; radiation²: t = 23.14, $P < 0.001$). Spatio-phenological components and drivers of dragonfly CL were consistent to alternative definitions of assemblages (Supplementary Table S1).

### Change in the phenological pattern of colour lightness over years
Independent polynomial models of the phenological pattern of CL for each of the 24 years were consistent in their shape and explained between 34 and 48 percent of CL variation (all $P < 0.001$; Fig. 4a, Supplementary Fig. S4, Table S2). Phenological CL patterns advanced over years by 3.6 days per decade for the day when CL turned lighter than expected by chance (CL > 0) (Fig. 4b; $F_{1,22} = 6.12$, $R^2 = 0.18$, $P = 0.021$), and by 3.8 days per decade for the day when CL peaked (Fig. 4b; $F_{1,22} = 11.25$, $R^2 = 0.31$, $P = 0.003$). The day when CL became darker than expected by chance (CL < 0) and the length of the period where CL was lighter than expected by chance did not change over the years (Fig. 4b; $F_{1,22} = 1.87$, $P = 0.185$ and $F_{1,22} = 1.34$, $P = 0.259$, respectively). The magnitude of the peak with maximum CL showed a non-significant positive tendency (Fig. 4a, $F_{1,22} = 2.58$, $R^2 = 0.06$, $P = 0.122$). Seasonal radiation patterns showed no directional changes over years in contrast to temperature, whose values over the season increased (Fig. S6).

## Discussion
Our results based on a high resolution and comprehensive dataset showed that colour lightness of dragonfly assemblages varied phenologically in close linkage with seasonal changes in solar radiation. Considering the known link between colour and thermoregulation in dragonflies, medium and dark colours would seemingly allow early and late flight season species to deal with corresponding intermediate and low radiation, respectively, while light colours enable mid-season species to thermoregulate well under the highest seasonal radiation conditions. Our study therefore supports a role of thermal melanism in the phenological diversity patterns of insects. As both body colour and phenological tracking responses of dragonfly species could be expected to be under related selection pressures for matching thermoregulatory performance to seasonal climate, they would presumably have been coupled via evolution. One of the few similar studies providing mechanistic understanding on the phenological patterns of insects showed that phenological changes of body sizes of wild bees in Catalonia (Spain) followed Bergmann´s rule, allowing larger species to deal with cold temperatures early and late in the year[39]. In our study, solar radiation was the primary factor associated with the variation of colour lightness, in line with its direct mechanistic effect on heat gain[21] and with previous studies on thermal melanism[23], although other studies also support the contribution of temperature[27] in combination with radiation[24,28].

The phenological operation of the TMH in dragonflies underlines the well-known dependency of adult odonates on thermoregulation[31,32]. However, we did not find any support for either the spatial or the phenological operation of the TMH in the suborder of the much smaller damselflies, which could be explained by the generally lower thermoregulatory requirements of small flying insects[36]. Smaller insect species have lower thoracic temperature requirements, possibly because flight is energetically less demanding for them[31,32]. Similarly, a different relationship between phenology and thermal physiology based on body size was found by Osorio in 2016[39], where only the large – more endothermic—bee species followed the predictions of Bergmann's rule that body size should increase at lower temperatures, but not the less thermoregulatory demanding small species. Based on our results,

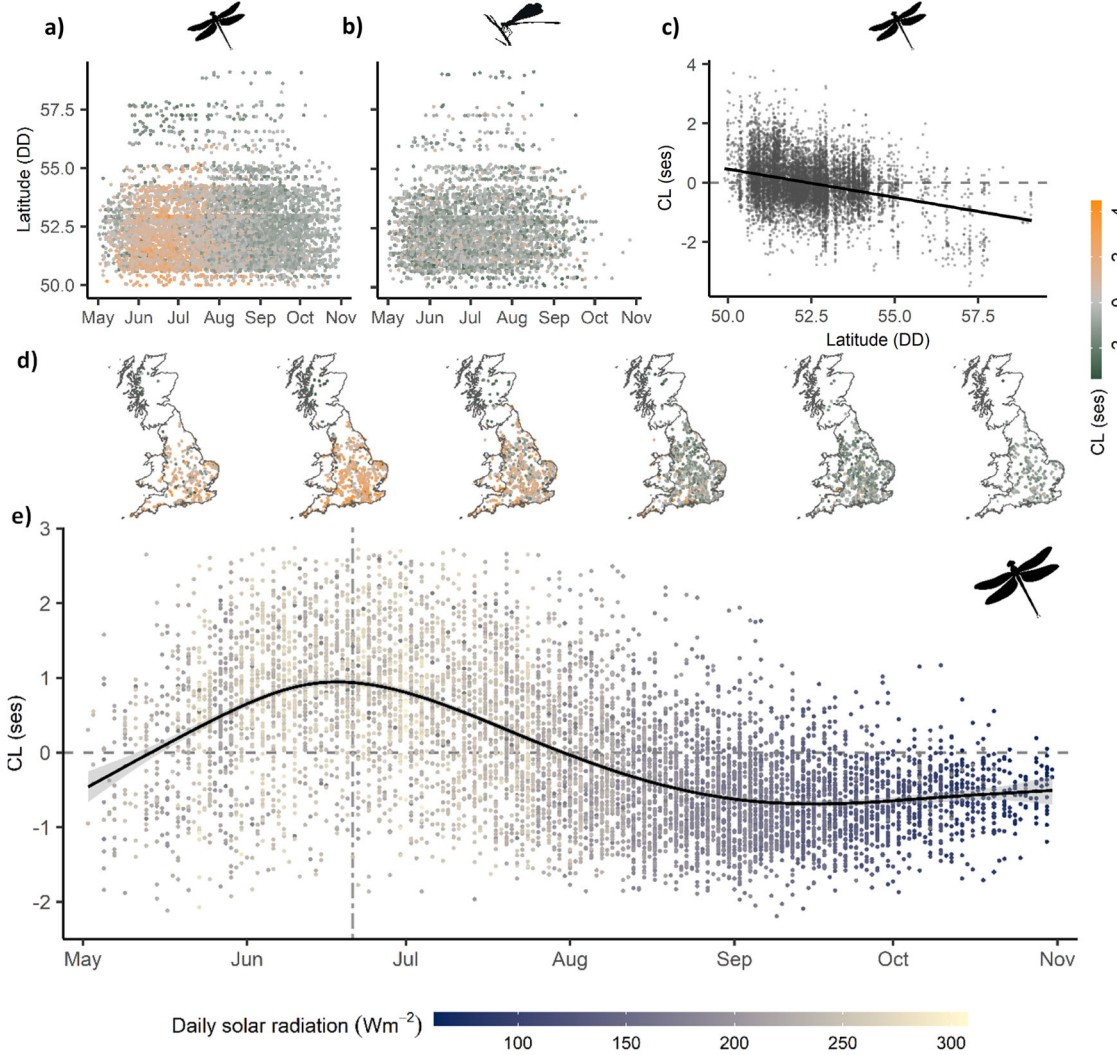

**Fig. 2 | Spatio-phenological variation of body colour lightness (CL) of dragonfly (a, c, d, e) and damselfly (b) assemblages in Great Britain.** Variation of CL of dragonfly (**a**) and damselfly (**b**) assemblages along latitude and season. **c** Residual CL variation of dragonfly assemblages with latitude, after removing the seasonal component. **e** Residual CL variation of dragonfly assemblages along the season after removing the latitudinal component. The black curve indicates LOESS regression with grey area indicating 95% confidence interval; point colour indicates solar radiation intensity received on the specific day of the year at the assemblage location; the vertical dashed line indicates summer solstice. CL is measured as community-weighted mean of body colour lightness in standard effect size units, i.e. as deviation of observed values from those of random assemblages (see "Methods"). CL values above zero (marked by the horizontal dashed line) indicate lighter assemblages than expected by chance, CL values below zero indicate darker assemblages than expected by chance.

previous assemblage-level support for TMH in odonates[27,28] may therefore be driven mostly by the dragonflies. Our contrasting results between taxonomically closely related taxa stresses the importance of accounting for the fundamental physiological differences of taxa in mechanistic ecological studies.

According to our analyses, the phenological pattern of dragonfly advanced over the last decades at rates of almost four days per decade, which aligns with previously reported advances of odonate flight periods of 1.5 days per decade in Great Britain[37], or 8.7 in the Netherlands[38]. In the latter, flight period length of species did not change, similar to our results where the length of the period above average CL did not change. Phenological advances of dragonfly flight periods imply that temperature, whose seasonal patterns are advancing over years[40], plays a primary role for determining dragonfly emergence, either by accelerating development[41] or as an environmental cue. This relegates solar radiation, whose seasonality is static over years[42], to play only a weak role as an environmental cue triggering dragonfly emergence even though our analysis suggest that it is the main factor linking to dragonfly flight periods.

Poor alignment between the cues triggering seasonal regulation and the drivers of optimal timing may likely prevent species from tracking shifting seasonal conditions[11,42] as it has been suggested in flowering plants[43]. Therefore, our study may suggest that phenological advances could desynchronize dragonfly flight periods from ideal seasonal conditions. For instance, earlier emergence of spring species may expose them to lower radiation than optimal based on body colour lightness, while summer species may be confronted with higher radiation than optimal. The complexity of the system makes, however, future prediction highly speculative. Furthermore, species´ may adjust to seasonal shifts by modifying their phenological responses to environmental cues[11] through either evolutionary adaptation or phenotypic plasticity mechanisms[44,45]. Similarly, species´ body colour lightness may respond to changes in climate or photoperiod, either evolutionarily[46] or via phenotypic plasticity[47,48]. The degree to which these mechanisms would be able to compensate observed and potential future phenological mismatches is, however, unknown (see[49,50]). More research on intraspecific colour variation, plasticity and adaptability in relation to seasonal environmental

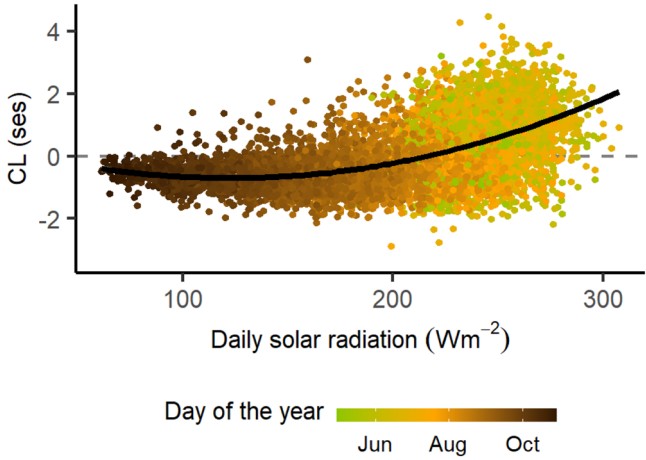

**Fig. 3 | Body colour lightness (CL) of dragonfly assemblages in relation to solar radiation.** CL is measured as community-weighted mean of body colour lightness in standard effect size units, i.e. as deviation of observed values from those of random assemblages (see "Methods"). Point colour indicates the day of the year during dragonfly flight season (May to October). Solar radiation is measured as the average solar radiation received on the specific day of the year at the assemblage location. The black curve indicates the regression line of a 2nd-degree polynomial model.

variation may offer clues about the potential of insect phenology to respond to a changing climate[11].

By providing evidence for a relationship between thermal melanism and ectotherm phenology, our study contributes to filling a gap in the comprehension of the essential but vastly understudied phenological component of diversity variation. Our results, which rely on fundamental mechanisms regulating species occurrences and life histories, may be representative for a broad spectrum of ectotherm taxa and stress the fundamental ecological importance of colour in driving diversity patterns of ectotherms. Note that while body colour is generally consistent within Odonata species, certain intraspecific variation may occur, and therefore our approach may not track fully colour lightness patterns. While the complexity of mechanisms driving phenology make generalizing predictions of the phenological component of the TMH a challenge, our results point to body colour as a key trait mediating the mechanisms and repercussions of recent phenological advances, which opens new research avenues to help elucidate general responses of species under global warming both in space and time[51]. We call for more phenological research that does not only report responses to triggering cues and phenological shifts, but addresses the mechanisms determining phenology, including the relative contributions of unregulated and regulated phenological tracking as well as mechanisms behind cue systems across taxa. Integrating the growing availability of massive high-resolution species' occurrence and environmental data together with increasingly comprehensive trait datasets opens opportunities for unprecedentedly detailed mechanistic insights into the spatio-temporal variation of diversity.

## Methods

### Building ecologically meaningful assemblages from occurrence records

We used the publicly available database of occurrence records (observations) of Odonata from the British Dragonfly Society Recording Scheme[33] (https://nbnatlas.org). We filtered the database by keeping records of adult individuals between 1990 and 2020. This resulted in 1,047,422 expert-validated observations of 56 species across Great Britain between May and October, i.e. during the flight season of odonates. We used these single-species observations to build multi-species assemblages separately for dragonflies (suborder Anisoptera) and damselflies (suborder Zygoptera). We define the spatio-

temporal dimensions and sampling representativeness of ecologically meaningful assemblages based on the following parameters, bearing in mind the definition of assemblage as a group of taxonomically related species that co-occur in space and time and are likely to interact[52,53]:

- *Spatial resolution (resSp)*, measured as the maximum distance between observations. Should be finer than the range of movements of observed individuals to allow them to have the potential to interact. We considered 0.1 km and 1 km for medium to large-size flying insects like odonates.
- *Phenological resolution (resPh)*, measured as the maximum difference in days of the year between observations. Should be fine relative to the patterns of phenological replacement of species to allow individuals observed within this timespan to have the potential to interact. We considered 7 and 14 days, finer than typical odonate flight period length (see Fig. 1).
- *Temporal resolution (resTem)*, measured as the maximum difference in years between observations. It allows for increasing sampling completeness. We considered conservative thresholds of 0 and 3 years to avoid compositional changes over years e.g. due to landcover or climate changes.
- *Sampling effort (samEf)*, measured as the number of sampling events. We chose a minimum of 4.
- *Sampling coverage (samCov)*, measured as the percentage of observed richness relative to the estimated richness from rarefaction curves (see below). We chose a conservative threshold of 80%.

We built assemblages by aggregating the database occurrence records within the spatio-temporal parameters *resSp*, *resPh*, *resTem*, for which we used the British national grid projection (EPSG:27700). To control for sampling representativeness, we then built, for each assemblage, species accumulation curves using the function *spaccum* (R package *vegan*[54]) and estimated predicted richness using the Chao index[55]. Those assemblages reaching the thresholds of *samEf* and *samCov* were kept and regarded as the ecologically meaningful assemblages that will be used. Species' presence/absence was used because abundance data was not systematically obtained in the occurrence record databases. Resulting point-based assemblages contain all key spatial, temporal and sampling representativity aspects of ecologically meaningful assemblages. This approach to build assemblages constitutes a step forward with respect to previous macroecological studies, which typically cluster observations at large spatial units[56], e.g.[57] or have poor control over sampling representativeness, e.g.[58,59], but see also[60]. Controlling for sampling representativeness reduces the likelihood of false absences resulting from insufficiently sampled assemblages, which may lead to misleading taxonomic and functional patterns[61].

We present our main results using the following parameters: *resSp* = 1 km, *resPh* = 14, *resTem* = 3, *samEff* = 4, *SamCov* = 80 aiming for a balance between fine spatio-temporal resolution and representative sample size. We obtained, with this parameter combination, 8159 dragonfly assemblages in 2570 locations having an average species richness of 4.06 +/− 1.95 (SD), and 4134 damselfly assemblages in 1613 locations (Fig. 2d) having an average species richness of 3.58 +/− 1.43. Our dataset contained 27 dragonfly and 19 damselfly species (Fig. 1d) representing all common odonate species in Great Britain. We also generated alternative sets of assemblages based on combinations of parameter values considered. Consistent results based on those are provided (Supplementary Table S1) to ensure that ecological patterns are robust to assemblage definition[62].

### Quantification of body colour lightness of assemblages

We measured body colour lightness of odonate species following an image-based analysis[27,28] in which we calculated the average of the

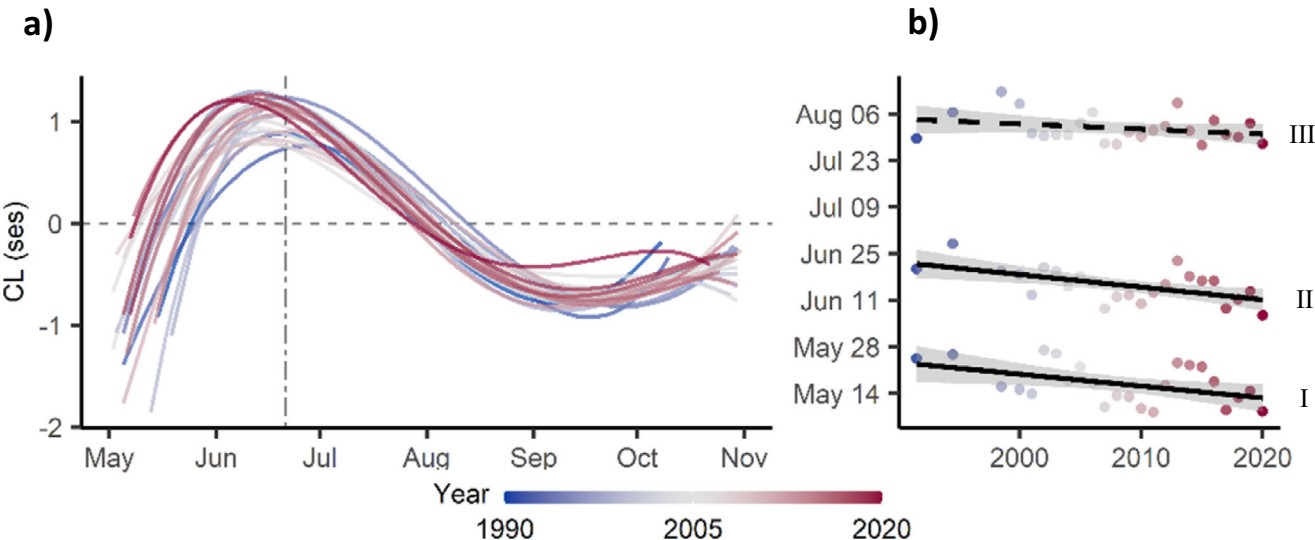

**Fig. 4 | Variation of the phenological pattern of body colour lightness (CL) of dragonfly assemblages between 1990 and 2020. a** Coloured lines represent regression lines of phenological models (4th-degree polynomial) of CL variation from 1990 (blue) to 2020 (red) (see Supplementary Fig. S4 for a depiction of the raw data for each year and Table S2 for statistical details); the vertical dashed line represents summer solstice. **b** Linear models showing the shift of attributes of phenological CL patterns over the 30-year period, specifically: (I) day of the year when CL turned lighter than expected by chance (CL > 0) ($P = 0.021$), (II) day of the year when CL peaked (maximum lightness) ($P = 0.003$), (III) day of the year when CL became darker than expected by chance (CL < 0) ($P = 0.185$); solid lines represent significant models ($P < 0.05$) and grey area 95% confidence intervals.

pixels of red, green and blue colour channels from scientific illustrations of individuals[34] and averaged them per species. This estimate of colour lightness has been confirmed to represent the physical ability of the species to absorb and reflect radiation energy as it was highly negatively correlated with the difference in species body temperature and ambient temperature ($r = -0.76$[28]). As male and female odonates often show different colouration, we focused on males for coherence with previous studies and because most odonatan observational records belong to males due to their more conspicuous behaviour. Note, however, that the colour lightness of sexes of the same species is highly correlated[28]. For each assemblage, we calculated community weighted means[35] of body colour lightness. We measured deviations of assemblage´s colour lightness from null expectations by randomising assemblage composition 100 times from the regional pool of species (all species in our dataset) and using the standard effect size[63]. Thereby we control for inherent biases of community weighted mean estimates[64]. All reported assemblage-level values of body colour lightness (hereafter simply CL) are standard effect sizes. Values of CL > 0 indicate higher colour lightness than expected by chance and values of CL < 0 indicate lower colour lightness than expected by chance.

We also calculated an alternative spatially-constrained CL measure that uses local instead of regional null models to measure standard effect sizes, which allows isolating the deviation in colour lightness corresponding to the phenological (i.e. non-spatial) replacement of species. For this, we divided the study area into quadrats of 100 km by 100 km, randomised the composition of assemblages based on the species pool within the respective quadrat and used those to quantify CL. For this analysis, we dropped the 326 and 321 assemblages of dragonflies and damselflies, respectively, that belonged to quadrats with less than 50 assemblages to ensure that species pools were in all cases representative. Results on spatially constrained CL are provided (Supplementary Fig. S1).

Species traits carry a phylogenetic signal that may lead to false interpretations of species trait-environment responses, particularly if phylogenetically related species show similar spatio-phenological patterns (see[65]). We used the most up-to-date molecular phylogeny of European odonates[66] and Lynch´s comparative method[67] to partition body colour lightness of species into a phylogenetic ($P$) and a species-specific ($S$) component[28]. $P$ represents the phylogenetically predicted part of trait variation, whereas $S$ represents the species-specific deviation from this phylogenetic prediction. We calculated CL separately for the $P$ and $S$ components of body colour (Supplementary Fig. S3). Consistent patterns in the $P$ and $S$ alternative measures of CL indicated robustness of our results against phylogenetic bias.

## Spatio-phenological variation of body colour lightness and its drivers

We quantified the magnitude of the variation of CL attributed to spatial and phenological dimensions on dragonflies by using a polynomial model with latitude and day of the year as predictor variables with polynomial terms to accommodate the curvilinear response of CL. We chose a 4th-degree polynomial based on model complexity and fit. For damselflies, CL did not show a curvilinear response to day of the year, therefore we used a linear term together with latitude. We present the latitudinal component of CL of dragonflies by using the residuals of a polynomial model of day of the year (Fig. 2c), and the phenological component (Fig. 2e) by using the residuals of a model with latitude. To evaluate the relative contribution of latitude and phenology to the variation in CL, we employed a hierarchical partitioning[68] analysis (R package *hier.part*). To determine whether CL patterns are a result of TMH, we analysed how much of the variation in CL was explained by environmental conditions of radiation and temperature. We downloaded, for each day of the year between 2004 and 2014, raster maps of surface downwelling shortwave radiation (rsds, hereafter radiation), which account for cloud cover, and near-surface air temperature (tas, hereafter temperature) (Chelsa dataset: w5e5v1.0[69] at 30 arcsec (~1 km) resolution). We averaged radiation and temperature values for each cell and day of the year across the 10-year period. We extracted values corresponding to the central sampling day and location of assemblages. Then, we used linear models to identify whether radiation or temperature drive the spatio-phenological variation in CL. We carried out model selection prioritising quadratic effects over interactions[70] and based on BIC criteria[71,72]. The final model included radiation with a quadratic effect and excluded temperature because it had minimal contribution (CL ~ radiation + radiation²). We used a semivariogram

(R package *gstat*[73]) to validate that spatial dependence in CL was weak (Fig. S5). Model assumptions were met in all models.

## Change of the phenological pattern of body colour lightness over years

To investigate whether and how the phenological CL pattern changed over the last three decades, we focused on the 97.3% of assemblages at latitudes below 55° N (Fig. 2a, b) to reduce variation of CL caused by latitude. We grouped records by year. Every year between 1990 and 1999 contained insufficient data to draw robust individual phenological patterns of CL (i.e. less than 80 days along the flight season). Therefore, we grouped together the years 1990–1993, 1994–1995, 1996–1997, and 1998–1999. For each of the 25 resulting year-groups of assemblages, we built individual polynomial models on variation of CL depending on day of the year, for which we used a 4th-degree polynomial (Fig. S4, Table S2). The model of year group 1996–1997 had much lower explanatory power ($R^2$: 0.18) than the rest ($R^2$: 0.34–0.48) due to poorly distributed assemblages over the season, and therefore we did not consider it. We characterised phenological patterns of CL for all other 24 models by extracting five attributes: day when CL turned lighter than expected by chance (CL > 0), day when CL peaked, day when CL became darker than expected by chance (CL < 0), number of days with lighter CL than expected by chance (CL > 0), and maximum CL. We tested whether these five attributes shifted over the period between 1990 and 2020 with linear models, using attributes as response variable and year as the predictor. We tested whether radiation and temperature have changed over the last decades in our study system.

We downloaded data[69] on temperature and radiation for each day of the year of every year between 1990 to 2016, and extracted the corresponding values of our assemblages. Then, we selected randomly 10 assemblage locations in the south and in the north of the study region. We represented, for either southern or northern groups of locations, changes in seasonal patterns of radiation and temperature across years (Fig. S6). We tested, with linear models, whether year had an effect in either temperature or radiation while accounting for the seasonal variation in those by using polynomial terms of day of the year: (radiation- Year +Day + $Day^2$ + $Day^3$, Temperature- Year + Day + $Day^2$ + $Day^3$).

### Reporting summary

Further information on research design is available in the Nature Portfolio Reporting Summary linked to this article.

## Data availability

The raw data on odonate observations and environment are available from https://nbnatlas.org and https://data.isimip.org with the identifiers https://doi.org/10.15468/cuyjyi and https://doi.org/10.48364/ISIMIP.836809.3, respectively. The processed datasets that support the findings of the study have been deposited to Zenodo https://doi.org/10.5281/zenodo.8003198.

## Code availability

The code necessary to run the analyses of this study has been made publicly available: https://doi.org/10.5281/zenodo.8003198.

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

## Acknowledgements

We thank the British Dragonfly Society (*BDS*) and its thousands of contributors for providing odonate occurrence data over 30 years, David Hepper from *BDS* for his assistance with the database as well as Benjamin Leroy and Matthew Biddick for their useful comments on the manuscript. This study was supported by the Bavarian Ministry of Science and the Arts via the Bavarian Climate Research Network Bayklif (project "mintbio"; R.N.F. and C.H.). S.P. acknowledges support by the Alexander-von-Humboldt Foundation.

## Author contributions

R.N.F.: Conceptualization, data curation and analyses, lead writing. S.P.: trait and phylogenetic data provision, manuscript revisions. D.Z.: trait data provision, manuscript revisions. R.B.: trait data provision, manuscript revisions. C.H.: Conceptualization, writing and manuscript revisions.

## Funding

## Competing interests

The authors declare no competing interests.
