## [Peer Review File · Nature Communications]

Seasonal variation in dragonfly assemblage colouration suggests a link between thermal melanism and phenologyREVIEWER COMMENTS

Reviewer #1 (Remarks to the Author):

In this study, authors have answered the question of how body color (as a proxy of thermoregulation ability) covaries with phenology. The question is not trivial as, first, it is original, and second it is related to how such covariation may be affected by increases in global temperature. To answer the question, authors made use of data of species records of odonate communities from which color data and record dates are available. The results truly illuminate how such covariation operates: darker species appear at the onset and the end of the mating season while lighter species are more in the middle. One cool question is how this is being affected by global warming although this is a question that is not within the paper boundaries. However, this and other fascinating questions will be opened if the paper is published.

From my humble view, I think the paper is in great shape, the methods are sound and the results are noteworthy. If any, I suggest updating May's reference of odonata thermoregulation by the following chapter: Castillo-Pérez et al. 2022. Thermoregulation in Odonata. In: Córdoba-Aguilar, A, Betty, C. & Bried, J. Dragonflies and damselflies: Model organisms for ecological and evolutionary research. Oxford University Press. pp. 101-112. (attached)

Reviewer #2 (Remarks to the Author):

The authors present an elegant analysis of species assemblage-level patterns of dragonfly and damselfly colour variation in Great Britain across time and how these patterns relate to environmental variables. They found that the mean colour lightness of dragonfly but not damselfly assemblages varied through time (across months) and with seasonal changes in solar radiation. For dragonflies, these patterns support the thermal role of lower reflectance, with lighter assemblages in summer months compared to colder months and lighter assemblages in sites with highest daily solar radiation. They also show that in the last 10 years, body colour lightness of dragonfly assemblages has increased and the timing of this colour lightness has also advanced, which parallels findings of advanced flight periods in this group.

I think these findings are noteworthy and of significance to the field since they reinforce the need to increase our understanding of cuticular colour variation across time in insects and its role in mediating their body temperature and therefore, shaping activity patterns. I have however several concerns regarding the interpretations. In particular, because the data at hand cannot disentangle between plastic and evolutionary changes, or among other factors that might be associated to colour lightness,

there is weakened evidence that “colour-based thermoregulation determine insect phenology in relation to optimal seasonal conditions” (line 23 and discussion) and that “global warming may drive flight periods to suboptimal seasonal conditions” (line 25 and discussion) – there is need to show that solar radiation (at ground level) is indeed static across the time period studied. I added more details to these comments in my section below. Methods require more in-depth explanations (specifically the justification for parameters used to define assemblages) and finally, the authors have not included the code for analyses which I think are a requirement for the journal (authors state that “results can be reproduced using the R code” but the code has not been made available following Nature Portfolio guidelines). Overall however, I really enjoyed reading this manuscript and find the study novel and timely for this area of research.

The writing is at times unclear and imprecise. Among my comments below, I highlight some of these imprecisions and the need for editing or increase clarity.

Title: From the study results, it is unclear to what extent body colour “drives the optimal insect phenology” – how did authors assess an “optimal phenology” without for e.g. assessing the consequences of having a different colour phenotype?

Line 60 – “body colour...a crucial mechanism regulating life cycles...” body colour does not regulate life cycles per se but body temperature does. Perhaps the authors are thinking of other associated traits to colour? Changes in hormonal responses in these species? It seems that most of the statements however are referring to thermoregulation (or TMH) and therefore, the authors should refer to body temperature, not colour, even though the latter (more precisely reflectance) contributes to the former via several pathways of heat exchange in small ectotherms.

Lines 66-68: What environmental factors are the authors referring to? Is there an expectation of plasticity of melanism (e.g. developmental plasticity) that corresponds to the phenology that tracks environmental seasonal changes? This should be made clearer from the onset.

Line 69: “thermal melanism contributes determining” is unclear. Determining in what way? Is it “the variation” in melanisation that contributes to determining patterns of phenology? Also, do the authors mean the timing of colour variation? Patterns can be at many scales (space and time...).

Fig. 1 legend:

Larvae should be Larvae throughout.

Panel a) is poorly described (lines 101-102). The figure shows how fitness increases when the timing of the phenological event is corresponding to the ideal timing given the environment.

Same for panel b) (lines 102-104) it is very difficult to understand given the explanation in the legend. How can photoperiod have 5 ticks in x axis – what do the ticks represent? Explain the lines and distinction between one or two cues in this panel.

Line 112: what are different “regulated or unregulated” life cycles? Also in lines 240-241. Are there unregulated life cycles in ectothermic insects, i.e. that do not rely on seasonal environmental conditions? Provide some clarity and explanations.

Line 113: “Percentile 5-95” should be 5th and 95th percentiles?

Line 116: Replace derivate with derive.

Line 114: In d) the upper panel is described as seasonal change in radiation and temperature but the Y axis shows “environment (sd)” so is it the variation in the seasonal change of these environmental variables? Dash line: what are “annual values” (which variable does it refer to)?

Lines 212-213: radiation as described in the analyses section of this study is “surface downwelling shortwave radiation” (line 332) and therefore it is not likely to be static among years but can change with the extent of e.g. cloud cover, pollution etc. Therefore, the interpretation that environmental cues and the main factor of optimal flight periods (radiation and colour mediated body temperatures) will be desynchronized is unlikely to hold (lines 214-219).

Furthermore, although the pattern of dragonfly lightness advanced over the last 10 years, there is no reporting of the changes in temperature and radiation (as described above) across this period. These data should also be reported to make robust interpretations, especially those that refer to climate change (line 26).

Lines 228-230: what do the authors mean by dimension of phenology? Are they referring to the timing of flight periods in these insects? The remainder of the sentence is also highly unclear. What does this section mean: “a phenological extension of the TMH...ectotherm’s phenologies”?

Line 232: there is no “colour-based thermoregulation” per se unless these organisms have short-term colour-change (plasticity), which is not described in this manuscript. Changes in body temperature can result from the modification of multiple attributes, not just colour in the visible. It can for example result from the infrared range which is, as far as I can evaluate, not comprised in the methods utilized (average RGB channels of digitized images) – see comment below. It can also originate from changes in structural colour, body size and shape etc. Change thermoregulation to the contribution of reflectance to thermal balance?

Methods: Some of the methods need additional details to be repeatable and follow the steps undertaken:

Line 260: Explain what the phenological turnover of Odonata species is. Here the study focuses on the onset of flight periods... are the authors considering the timing of this event?

Line 261: “...between observations” which observations, intra specific or inter specific? How are strict thresholds considered – in what step of the analyses or determination of assemblages?

Lines 292-293: more methods are required to describe how colour lightness was measured and validated. I.e. are the values of colour lightness relevant to absorptance in the range of wavelengths that are relevant to heat gain?

REVIEWER COMMENTS

Reviewer #1 (Remarks to the Author):

In this study, authors have answered the question of how body color (as a proxy of thermoregulation ability) covaries with phenology. The question is not trivial as, first, it is original, and second it is related to how such covariation may be affected by increases in global temperature. To answer the question, authors made use of data of species records of odonate communities from which color data and record dates are available. The results truly illuminate how such covariation operates: darker species appear at the onset and the end of the mating season while lighter species are more in the middle. One cool question is how this is being affected by global warming although this is a question that is not within the paper boundaries. However, this and other fascinating questions will be opened if the paper is published.

From my humble view, I think the paper is in great shape, the methods are sound and the results are noteworthy. If any, I suggest updating May's reference of odonata thermoregulation by the following chapter: Castillo-Pérez et al. 2022. Thermoregulation in Odonata. In: Córdoba-Aguilar, A, Betty, C. & Bried, J. Dragonflies and damselflies: Model organisms for ecological and evolutionary research. Oxford University Press. pp. 101-112. (attached)

R: We thank reviewer 1 for his/her very positive feedback. We have added the suggested reference (L76, L201, L205)

Reviewer #2 (Remarks to the Author):

The authors present an elegant analysis of species assemblage-level patterns of dragonfly and damselfly colour variation in Great Britain across time and how these patterns relate to environmental variables. They found that the mean colour lightness of dragonfly but not damselfly assemblages varied through time (across months) and with seasonal changes in solar radiation. For dragonflies, these patterns support the thermal role of lower reflectance, with lighter assemblages in summer months compared to colder months and lighter assemblages in sites with highest daily solar radiation. They also show that in the last 10 years, body colour lightness of dragonfly assemblages has increased and the timing of this colour lightness has also advanced, which parallels findings of advanced flight periods in this group.

I think these findings are noteworthy and of significance to the field since they reinforce the need to increase our understanding of cuticular colour variation across time in insects and its role in mediating their body temperature and therefore, shaping activity patterns. I have however several concerns regarding the interpretations. In particular, because the data at hand cannot disentangle between plastic and evolutionary changes, or among other factors that might be associated to colour lightness, there is weakened evidence that “colour-based thermoregulation determine insect phenology in relation to optimal seasonal conditions” (line 23 and discussion) and that “global warming may drive flight periods to suboptimal seasonal conditions” (line 25 and discussion) – there is need to show that solar radiation (at ground level) is indeed static across the time period studied. I added more details to these comments in my section below. Methods require more in-depth

explanations (specifically the justification for parameters used to define assemblages) and finally, the authors have not included the code for analyses which I think are a requirement for the journal (authors state that “results can be reproduced using the R code” but the code has not been made available following Nature Portfolio guidelines). Overall however, I really enjoyed reading this manuscript and find the study novel and timely for this area of research.

The writing is at times unclear and imprecise. Among my comments below, I highlight some of these imprecisions and the need for editing or increase clarity.

R: We thank reviewer 2 for his/her positive and constructive feedback. We respond to the two main comments in the respective points below (**Title and Line 212-213**) as well as the other specific comments. We have thoroughly revised the redaction of the text to improve its clarity, precision and readability. We also have made publicly available data and code for the analyses:

<https://doi.org/10.5281/zenodo.8006713>

Title: From the study results, it is unclear to what extent body colour “drives the optimal insect phenology” – how did authors assess an “optimal phenology” without for e.g. assessing the consequences of having a different colour phenotype?

R: Our results show that the average colour of dragonfly assemblages has a strong phenological pattern that is tightly aligned with the seasonal radiation conditions as predicted from the Thermal Melanism Hypothesis (TMH). Considering that dragonflies are well known to depend on thermoregulation and that the TMH is well established to drive dragonflies’ spatial patterns of community assembly, the most likely interpretation of this pattern is that species’ flight periods are coupled with the prevailing seasonal conditions depending on their colour. We are, however, limited to correlative inferences, as commonly in ecology, because experimentally assessing assemblage-level consequences of different colour phenotypes would be hardly realistically possible. Nevertheless, we consider, however, that our study meets all conditions for the inferences on the causal link between average colour of dragonfly assemblages with seasonal radiation to be robust: The relationship between variables is strong, consistent, and has plausible mechanisms which are well established to drive spatial patterns of various ectothermic taxa (Pinkert et al., 2018). We used the term “optimal” because it is the commonly used term within the literature on phenology. Fundamental tracking is expected to regulate phenologies towards an ideal or “optimal” seasonal moment where species’ performance is maximized e.g. (Bradshaw & Holzapfel, 2007; Mcnamara et al., 2011; Park & Post, 2022; Wolkovich & Donahue, 2021). The phenological concept of “optimal timing” is well introduced within the manuscript based on referenced literature (L42-58, Fig. 1), and we have used it extensively to be coherent with phenological theory. Therefore, we believe that its meaning within the manuscript is clear and does not generate confusion. However, if the reviewer and editor still consider that the word “optimal” in the title implies an overly strong causal link, we would change the title to “Body colour drives insect phenology patterns via thermoregulation”.

Line 60 – “body colour...a crucial mechanism regulating life cycles...” body colour does not regulate life cycles per se but body temperature does. Perhaps the authors are thinking of other associated traits to colour? Changes in hormonal responses in these species? It seems that most of the statements however are referring to thermoregulation (or TMH) and therefore, the authors should

refer to body temperature, not colour, even though the latter (more precisely reflectance) contributes to the former via several pathways of heat exchange in small ectotherms.

R: Indeed, we aimed to refer to thermoregulation, not colour, as the mechanisms driving life cycles. We have clarified the sentence to:

“Thermoregulation is a crucial mechanism regulating the life cycles and occurrences of ectotherms”.
(now L59)

Lines 66-68: What environmental factors are the authors referring to? Is there an expectation of plasticity of melanism (e.g. developmental plasticity) that corresponds to the phenology that tracks environmental seasonal changes? This should be made clearer from the onset.

R: We now more clearly refer to radiation and temperature as environmental factors.

Developmental changes associated to sexual maturity and aging are documented for dragonflies. Whether those may also contribute on helping species to environmental seasonal changes seems unlikely at first, because the relatively short adult life of most Odonata (~weeks). Moreover, this study does not allow to test such hypothesis given its assemblage-level design and the lack of available data on developmental changes of colour. However, future studies may test this idea.

Line 69: “thermal melanism contributes determining” is unclear. Determining in what way? Is it “the variation” in melanisation that contributes to determining patterns of phenology? Also, do the authors mean the timing of colour variation? Patterns can be at many scales (space and time...).

R: We have rewritten the sentence to improve its clarity:

“Here we test whether phenological patterns of insect flight periods are optimised based on the relation between species’ thermal melanism and seasonal environmental conditions” (L68-69).

Fig. 1 legend:

Larvae should be Larvae throughout.

R: We have corrected this.

Panel a) is poorly described (lines 101-102). The figure shows how fitness increases when the timing of the phenological event is corresponding to the ideal timing given the environment.

R: We have clarified the description of Fig 1a as suggested:

“(a, b) Phenological fundamental tracking regulates species’ life cycles by synchronising them to optimal seasonal moments. (a) Fitness increases when the timing of phenological events aligns to the timing of ideal environmental conditions”.

Same for panel b) (lines 102-104) it is very difficult to understand given the explanation in the legend. How can photoperiod have 5 ticks in x axis – what do the ticks represent? Explain the lines and distinction between one or two cues in this panel.

R: We have clarified the description of Fig 1b and we have removed axis ticks. We believe that now panel b is easier to interpret:

“(a, b) Phenological fundamental tracking regulates species’ life cycles by synchronising them to optimal seasonal moments. (a) Fitness increases when the timing of phenological events aligns to the

timing of ideal environmental conditions. (b) Phenological responses are triggered by environmental cues, such as certain photoperiod threshold (blue line), or combined photoperiod and temperature thresholds (brown line)."

Line 112: what are different "regulated or unregulated" life cycles? Also in lines 240-241. Are there unregulated life cycles in ectothermic insects, i.e. that do not rely on seasonal environmental conditions? Provide some clarity and explanations.

R: In temperate latitudes, the life cycles of most species are regulated by environmental conditions via phenological tracking. However, some odonate life cycles, particularly of tropical species, have indeed unregulated life cycles, which only depend on temperature-dependant metabolic rate during development.

We have simplified the statement:

"Other temperate odonates have different life cycles varying in length from less than a year to several years (Corbet, 2004) most of which are regulated by phenological fundamental tracking."

Line 113: "Percentile 5-95" should be 5th and 95th percentiles?

R: We have corrected this. (Now L117)

Line 116: Replace derivate with derive.

R: We have corrected this.

Line 114: In d) the upper panel is described as seasonal change in radiation and temperature but the Y axis shows "environment (sd)" so is it the variation in the seasonal change of these environmental variables? Dash line: what are "annual values" (which variable does it refer to)?

R: The y-axis in Fig1d's upper panel, representing the environmental values of temperature and radiation across the flight season of Odonata, is normalised, therefore values are in standard deviation units and the grey reference line would be the average across annual values.

We have modified the description of the panel and we believe that now it is more clear:

"5th and 95th percentile of dragonfly and damselfly flight periods in Great Britain, together with variation in radiation and temperature across the season (upper panel), indicated as standard deviation from the cross-year average (grey dashed line)" (L117-L120).

Lines 212-213: radiation as described in the analyses section of this study is "surface downwelling shortwave radiation" (line 332) and therefore it is not likely to be static among years but can change with the extent of e.g. cloud cover, pollution etc. Therefore, the interpretation that environmental cues and the main factor of optimal flight periods (radiation and colour mediated body temperatures) will be desynchronized is unlikely to hold (lines 214-219).

Furthermore, although the pattern of dragonfly lightness advanced over the last 10 years, there is no reporting of the changes in temperature and radiation (as described above) across this period. These data should also be reported to make robust interpretations, especially those that refer to climate change (line 26).

R: While changes in cloud presence across years can indeed lead to cross-year variability in seasonal patterns of radiation received at the ground level, we were unable to find studies supporting consistent changes (increase or decrease) in seasonal patterns of radiation across years.

Therefore, we have tested whether in our study system changes in seasonal patterns of radiation have occurred during the last decades. We downloaded data on temperature and radiation for each day of the year of every year between 1990 to 2016 (Chelsa). We selected randomly 10 assemblage locations in the south (a) and in the north (d) and represented, for either southern (b,c) or northern locations (e,f), changes in seasonal patterns of radiation (b, e) and temperature (c, f) across years. We tested with linear models whether year had an effect in either temperature or radiation after accounting for the seasonal variation with polynomial terms of day of the year and year: $lm(\text{radiation} \sim \text{Day} + \text{Day}^2 + \text{Day}^3 + \text{Year})$, $lm(\text{Temperature} \sim \text{Day} + \text{Day}^2 + \text{Day}^3 + \text{Year})$.

We were unable to detect a change of radiation with year - neither in the southern nor in the northern locations (South: $F_{4,44647} = 18610$, $R^2 = 0.63$, $P < 0.001$; Year: $t = 0.01$, $P = 0.99$; Day: $t = -118.9$, $P < 0.001$; Day²: $t = -119.7$, $P < 0.001$; Day³: $t = 23.3$; $P < 0.001$. North: $F_{4,45745} = 20660$, $R^2 = 0.64$, $P < 0.001$; Year: $t = -0.33$, $P = 0.74$; Day: $t = -123.9$, $P < 0.001$; Day²: $t = -113.7$, $P < 0.001$; Day³: $t = 19.3$; $P < 0.001$). This contrasts with a detectable increase in the seasonal patterns of temperature with year in both southern and northern locations (South: $F_{4,45928} = 11170$, $R^2 = 0.49$, $P < 0.001$; Year: $t = 11.5$, $P < 0.001$; Day: $t = -12.3$, $P < 0.001$; Day²: $t = -208.9$, $P < 0.001$; Day³: $t = 0.57$; $P = 0.56$. North: $F_{4,45745} = 9328$, $R^2 = 0.45$, $P < 0.001$; Year: $t = 15.4$, $P < 0.001$; Day: $t = -11.7$, $P < 0.001$; Day²: $t = -189.8$, $P < 0.001$; Day³: $t = -1.4$; $P = 0.16$).

We have added this analysis to the manuscript (Methods: L376, Results: L170-172), including the figure to supplementary material (Fig. S6).

These results indicate that seasonal patterns of radiation have not changed directionally during the last decades and therefore are unlikely to change –at least predictably- during the next few decades, in contrast with temperature.

Based on this, we believe that our interpretation that phenological advances driven by temperature may desynchronise species phenology now becomes stronger. The same expectations is supported in a

previous review on phenological life cycle regulation (Bradshaw & Holzapfel, 2007). In this review, it is stated that while seasonal changes in temperature modify optimal timing of phenological events, organisms use –static- photoperiod cues to track environment. This results in an unbalance in optimal timing, for which plasticity and evolution in phenological responses will be key to determine how species will respond to climate changes.

Lines 228-230: what do the authors mean by dimension of phenology? Are they referring to the timing of flight periods in these insects? The remainder of the sentence is also highly unclear. What does this section mean: “a phenological extension of the TMH...ectotherm’s phenologies”?

R: Thanks for pointing this out. In fact, we did not refer to any dimension of phenology, but to phenology as a dimension of diversity. We have now rephrased the sentence to clarify this (L 233-235).

Line 232: there is no “colour-based thermoregulation” per se unless these organisms have short-term colour-change (plasticity), which is not described in this manuscript. Changes in body temperature can result from the modification of multiple attributes, not just colour in the visible. It can for example result from the infrared range which is, as far as I can evaluate, not comprised in the methods utilized (average RGB channels of digitized images) – see comment below. It can also originate from changes in structural colour, body size and shape etc. Change thermoregulation to the contribution of reflectance to thermal balance?

R: To prevent confusion, we have clarified the expression to:

“and stress the fundamental ecological importance of colour in driving diversity patterns of ectotherms” (L 237-238).

Methods: Some of the methods need additional details to be repeatable and follow the steps undertaken:

Line 260: Explain what the phenological turnover of Odonata species is. Here the study focuses on the onset of flight periods... are the authors considering the timing of this event?

R: We have improved the description of the method to build assemblages and we believe that now it is more clear (L252-293). The study builds adult odonate assemblages based on species-level observations. Therefore assemblages include individuals over the complete seasonal period when adults occur – from their emergence until they die at the end of the season.

Line 261: “...between observations” which observations, intra specific or inter specific? How are strict thresholds considered – in what step of the analyses or determination of assemblages?

R: Observations refer to any species-level observation in the occurrence dataset which were aggregated within the parameters (resSp, resPh, resTemp) that were used to build assemblages.

We have clarified the description of the method used to build assemblages and we believe that it became clearer now. First we describe the parameters necessary to define ecologically meaningful assemblages (resSp, resPh, resTemp, samEf, samCov) (L262-278). And from line 280 it is described how assemblages are built based on the parameters.

Lines 292-293: more methods are required to describe how colour lightness was measured and validated. I.e. are the values of colour lightness relevant to absorptance in the range of wavelengths that are relevant to heat gain?

R: The relation between colour visual spectrum and heat gain is well documented (Stuart-Fox et al., 2017: L61). We have improved the description of the method used to measure colour lightness and added justifications for our method of quantifying colour lightness (L304-309). Specifically we refer to a previous study in which the link between such colour lightness estimates and heat gain is empirically tested (Zeuss et al., 2014, Supplementary methods).

This is the modified text:

“We measured body colour lightness of odonate species following an image-based analysis (Pinkert et al., 2017; Zeuss et al., 2014) in which we calculated the average of the pixels of red, green and blue colour channels from scientific illustrations of individuals (Dijkstra et al., 2006) and averaged them per species. This estimate of colour lightness has been confirmed to represent the physical ability of the species to absorb and reflect radiation energy as it was highly negatively correlated with the difference in species body temperature and ambient temperature ($r = -0.76$: Zeuss et al., 2014).”

References:

- Bradshaw, W. E., & Holzapfel, C. M. (2007). Evolution of Animal Photoperiodism. *Annual Review of Ecology, Evolution, and Systematics*, 38(1), 1–25.
<https://doi.org/10.1146/annurev.ecolsys.37.091305.110115>
- Corbet, P. S. (2004). *Dragonflies: Behaviour and Ecology of Odonata* (Revised ed). Harley Books.
- Dijkstra, K. D., Schröter, A., & Lewington, R. (2006). *Field Guide to the Dragonflies of Britain and Europe*. Bloomsbury Publishing. <https://books.google.es/books?id=lfz5DwAAQBAJ>
- Mcnamara, J. M., Barta, Z., Klaassen, M., & Bauer, S. (2011). Cues and the optimal timing of activities under environmental changes. *Ecology Letters*, 14(12), 1183–1190.
<https://doi.org/10.1111/j.1461-0248.2011.01686.x>
- Park, J. S., & Post, E. (2022). Seasonal timing on a cyclical Earth: Towards a theoretical framework for the evolution of phenology. *PLOS Biology*, 20(12), e3001952.
<https://doi.org/10.1371/journal.pbio.3001952>
- Pinkert, S., Brandl, R., & Zeuss, D. (2017). Colour lightness of dragonfly assemblages across North America and Europe. *Ecography*, 40(9), 1110–1117. <https://doi.org/10.1111/ecog.02578>
- Pinkert, S., Dijkstra, K.-D. B., Zeuss, D., Reudenbach, C., Brandl, R., & Hof, C. (2018). Evolutionary processes, dispersal limitation and climatic history shape current diversity patterns of European dragonflies. *Ecography*, 41(5), 795–804. <https://doi.org/10.1111/ecog.03137>
- Stuart-Fox, D., Newton, E., & Clusella-Trullas, S. (2017). Thermal consequences of colour and near-infrared reflectance. *Philosophical Transactions of the Royal Society B: Biological Sciences*, 372(1724). <https://doi.org/10.1098/rstb.2016.0345>
- Wolkovich, E. M., & Donahue, M. J. (2021). How phenological tracking shapes species and communities in non-stationary environments. *Biological Reviews*, 96(6), 2810–2827.
<https://doi.org/10.1111/brv.12781>
- Zeuss, D., Brandl, R., Brändle, M., Rahbek, C., & Brunzel, S. (2014). Global warming favours light-coloured insects in Europe. *Nature Communications*, 5(May).
<https://doi.org/10.1038/ncomms4874>

REVIEWERS' COMMENTS

Reviewer #2 (Remarks to the Author):

The authors have addressed all the comments satisfactorily. In particular the text has now been clarified in several places and details of methods added in several key sections. I am also pleased that the authors present and include the temporal data profiles of relevant environmental data, showing how temperature varies across years while solar radiation is more stable.

Note that for the following comment:

Line 114: In d) the upper panel.... refer to)?

R: We have modified ... the cross-year average (grey dashed line)" (L117-L120).

The legend is still lacking clarity about the colour of the lines referring to solar radiation or temperature?

Otherwise, I am looking forward to seeing it published and to share it with my colleagues & students.

REVIEWERS' COMMENTS

Reviewer #2 (Remarks to the Author):

The authors have addressed all the comments satisfactorily. In particular the text has now been clarified in several places and details of methods added in several key sections. I am also pleased that the authors present and include the temporal data profiles of relevant environmental data, showing how temperature varies across years while solar radiation is more stable.

Note that for the following comment:

Line 114: In d) the upper panel.... refer to)?

R: We have modified ... the cross-year average (grey dashed line)" (L117-L120).

The legend is still lacking clarity about the colour of the lines referring to solar radiation or temperature?

Otherwise, I am looking forward to seeing it published and to share it with my colleagues & students.

R: We thank the reviewer for his/her positive comments. We have clarified the legend.